# A CRISPR/Cas9-based enhancement of high-throughput single-cell transcriptomics

Amitabh C. Pandey [1,2,3,9] ✉, Jon Bezney[4,5,6,9], Dante DeAscanis[5,9], Ethan B. Kirsch[3], Farin Ahmed[4], Austin Crinklaw [5], Kumari Sonal Choudhary[5], Tony Mandala[4], Jeffrey Deason[5], Jasmin S. Hamidi[3], Azeem Siddique[5], Sridhar Ranganathan[5], Keith Brown [5], Jon Armstrong[5], Steven Head[4], Phillip Ordoukhanian[4], Lars M. Steinmetz [6,7,8] & Eric J. Topol [3]

Single-cell RNA-seq (scRNAseq) struggles to capture the cellular heterogeneity of transcripts within individual cells due to the prevalence of highly abundant and ubiquitous transcripts, which can obscure the detection of biologically distinct transcripts expressed up to several orders of magnitude lower levels. To address this challenge, here we introduce single-cell CRISPRclean (scCLEAN), a molecular method that globally recomposes scRNAseq libraries, providing a benefit that cannot be recapitulated with deeper sequencing. scCLEAN utilizes the programmability of CRISPR/Cas9 to target and remove less than 1% of the transcriptome while redistributing approximately half of reads, shifting the focus toward less abundant transcripts. We experimentally apply scCLEAN to both heterogeneous immune cells and homogenous vascular smooth muscle cells to demonstrate its ability to uncover biological signatures in different biological contexts. We further emphasize scCLEAN's versatility by applying it to a third-generation sequencing method, single-cell MAS-Seq, to increase transcript-level detection and discovery. Here we show the possible utility of scCLEAN across a wide array of human tissues and cell types, indicating which contexts this technology proves beneficial and those in which its application is not advisable.

While scRNAseq has revolutionized the analysis of individual cells, challenges with handling data sparsity only continue to rise[1]. Such technical sources of data sparsity are contributed by batch effects, input material quantity variation, and amplification bias[2–9]. Additionally, limitations in conversion efficiency from mRNA to cDNA allow only a small fraction of the mature polyadenylated (polyA) mRNA to be captured[10–14], potentially confounding biological interpretation and reproducibility.

Experimental approaches are continually developed to combat this inherent limitation[11,15–18].

However, existing methods continue to face the same challenges of data sparsity wherein, regardless of capture method, molecules are sampled from the same underlying distribution that favors moderately to highly abundant polyA transcripts[19]. This means that each new sequencing read has the highest probability of representing a highly expressed transcript and its artifactual duplicates (PCR or optical).

[1]Section of Cardiology, Tulane Heart and Vascular Institute, Department of Medicine, Tulane University School of Medicine, New Orleans, LA, USA. [2]Southeast Louisiana Veterans Health Care System, New Orleans, LA, USA. [3]Department of Molecular Medicine, Scripps Research Translational Institute, The Scripps Research Institute, La Jolla, CA, USA. [4]Genomics Core Facility, The Scripps Research Institute, La Jolla, CA, USA. [5]Jumpcode Genomics, San Diego, CA, USA. [6]Department of Genetics, Stanford University School of Medicine, Stanford, CA, USA. [7]Stanford Genome Technology Center, Palo Alto, CA, USA. [8]European Molecular Biology Laboratory (EMBL), Genome Biology Unit, Heidelberg, Germany. [9]These authors contributed equally: Amitabh C. Pandey, Jon Bezney, Dante DeAscanis. ✉e-mail: apandey@tulane.edu

Whereas, if the highly abundant molecules are removed from the library prior to sequencing, then each new sequencing read is sampled from a distribution enriched for lesser abundant molecules.

Here, we present single-cell CRISPRclean (scCLEAN), a method that can be applied to any existing scRNAseq library with a dsDNA-intermediate. scCLEAN utilizes CRISPR/Cas9 to selectively cut and remove highly abundant molecules to redistribute sequencing reads to lesser abundant molecules. While CRISPR/Cas9 has been previously integrated into NGS methods to excise unwanted rRNA sequences[20–23], we demonstrate the adaptability of scCLEAN in an additional application by adapting it for a droplet-based polyA-captured scRNAseq method. scCLEAN selectively removes multiple categories of uninformative molecules (both genomic and transcriptomic), not limited to rRNA. Given that single-cell analysis hinges on detecting differences in gene expression between cells, we propose that adjusting the distribution of the sequencing library towards less ubiquitous transcripts would improve the discovery of biologically distinct relationships between cells. Essentially, biologically relevant transcripts that were previously discarded due to falling below the noise threshold would now be detectable and contribute significantly to enhancing the overall signal-to-noise ratio.

We leverage in silico modeling of diverse publicly available scRNAseq datasets (generated with the 10X Genomics 3′ v3.1 method) to identify genetic targets with both high relative abundance and minimal variance across all human tissue types. We further perform extensive characterization detailing the suitability of the technology across human cell types and tissues. Thereafter, we experimentally validate scCLEAN on a well characterized heterogeneous sample type, peripheral blood mononuclear cells (PBMCs), which partly suffer from data sparsity and high noise due to low mRNA content of immune cell types[24]. The impact of scCLEAN on single-cell analysis is then rigorously examined using multiple orthogonal workflows, comparing scCLEAN to in silico depletion analysis and incorporating into reference atlases to showcase its integration into existing data. Finally, to highlight the ability of scCLEAN to enhance biological hypothesis generation, we apply the method to multiple biological replicates of primary vascular smooth muscle cells (VSMCs) isolated from the coronary and pulmonary artery of healthy matched donors. Using scCLEAN, we uncovered inflammatory signatures within the coronary artery, shedding light on the pathogenesis of coronary artery disease.

## Results

### scCLEAN is applicable across human cell types and tissues

To establish a framework of candidate molecules that could safely be removed from a polyA-capture scRNAseq library without influencing downstream analysis, we focused on the 10x Genomics Chromium platform due to its widespread adoption and prevalence[25]. We surveyed 14 tissue types across 14 different 10X Genomics studies utilizing the 3′ v3.1 method to underscore highly abundant and unwanted sequences to be targeted by scCLEAN (Fig. 1a, Supplementary Data 1). We initially highlighted abundantly covered genomic regions lacking transcript annotations (see "Methods"), as these intergenic regions are filtered out before scRNAseq data processing. We also detected rRNA-aligned reads, despite using a polyA capture method to remove rRNA. Thus, we also rationalized that carryover of rRNA is deserving as candidates for removal given their prevalence in the queried datasets. Taken together, these two sources contributed to a median of 10% and 9% of the total aligned read count, respectively (Fig. 1b). In addition, we unsurprisingly found a high fraction of transcriptomic reads aligning to mitochondrially-encoded genes and nuclear-encoded ribosomal protein-coding genes. We highlighted 10 mitochondrially-encoded genes and 90 nuclear-encoded ribosomal genes common to all surveyed studies (further referred to as Ribo/Mito) that together constitute a median of 34% of total aligned reads (Fig. 1b). Together,

mitochondrial and ribosomal genes have been documented to contribute to a high proportion of counts in single-cell data across multiple methods[26]. From there, we also rationalized that other highly abundant and ubiquitous protein-coding genes are also obscuring downstream biological interpretation based on limited biological variability.

To assess highly abundant and low-variance genes, we leveraged the Pegasus python toolkit to conservatively filter for targetable genes sharing a combination of both high ranked mean expression and low variance across all 14 tissue types (see "Methods"). From there, 155 protein-coding genes (Supplementary Data 2), further referred to as non-variable genes (NVGs) due to their low variance ranked by mean UMI counts across all tissues, were curated. The NVGs were found to contribute to a median of 5% of non-deduplicated sequencing reads (Fig. 1b). Thus, we concluded that ~58% of median reads can be redistributed to the remaining transcriptome by removing genomic-derived intervals, rRNAs, and 0.7% of the transcriptome (Ribo/Mito and NVGs) (Supplementary Data 3).

To further characterize the ubiquitousness of the 255 protein-coding genes (Ribo/Mito and NVGs) selected for removal, we pursued accurate parameter optimization by querying the Tabula Sapiens single-cell atlas[27]. Across all 161 cell types, the 255 targeted genes were collectively in the lowest bins of gene variability and the highest bins of gene expression (Fig. 1c, d)[28]. Additionally, the 255 targets were confirmed to be negatively associated with variability when examining through gene-set enrichment analysis (GSEA). (Fig. 1e, Supplementary Fig. 1). According to the Tabula Sapiens dataset, we identified 8 targeted genes from the 255 gene panel (MIR205HG, CSTA, S100A2, KRT5, RPS27, B2M, FTL, MT-CO3) that might confer cell type specific expression (normalized variance > 1) (Fig. 1f) depending on cellular context.

After characterizing the 255 gene panel for ubiquitousness in the Tabula Sapiens dataset, we sought to generate a comprehensive metric to assess the impact of the 255 gene panel on any human tissue of interest. To do so, we queried all tissue types stored in The Genotype-Tissue Expression (GTEx) project and quantified the normalized gene expression in transcripts per million (TPM) for the targeted 255 gene panel. From there, we measured that ~50% of total TPM from heart, brain, and kidney tissues were contributed by the scCLEAN-targeted gene panel, while whole blood had the lowest with <20% total TPM (Supplementary Fig. 2). This provides a gauge to which tissue types scCLEAN would have the most potential benefit. Conversely, to further quantify a potential loss in biological marker genes across all queried tissues, genes with tissue specific expression (extended tau score and differential expression) within the 255 gene panel were tallied[29,30]. While most tissues (87.5%) resulted in no removal of marker genes, whole blood was highlighted as the least applicable tissue with 4 marker genes targeted for removal (Fig. 1g, Supplementary Fig. 2).

To address the potential concern of the targeted marker genes at the single-cell level, we again leveraged the Tabula Sapiens dataset to highlight erythrocyte and erythroid lineage cell types with the least predicted added value from scCLEAN (Supplementary Fig. 3)[27]. Furthermore, cell types ($n = 1598$) with distinctive gene markers analyzed via the CellMarker 2.0 database revealed only 0.44% of cell types ($n = 7$) contained a marker gene in the 255 gene panel (Supplementary Fig. 4)[31]. Furthermore, the cell type with the highest percent of total markers removed, exhausted CD8 + T cell, retained ~94% of total cell markers. Of the 7 cell-types containing marker genes found in the scCLEAN panel, 4 were immune-related cells isolated from blood, corroborating results from the tissue level analysis (Supplementary Fig. 4). To experimentally validate the applicability of scCLEAN despite the removal of predicted marker genes relevant to immune-related cell types, we first performed proof-of-concept experiments on ~40,000 PBMCs isolated from a healthy donor.

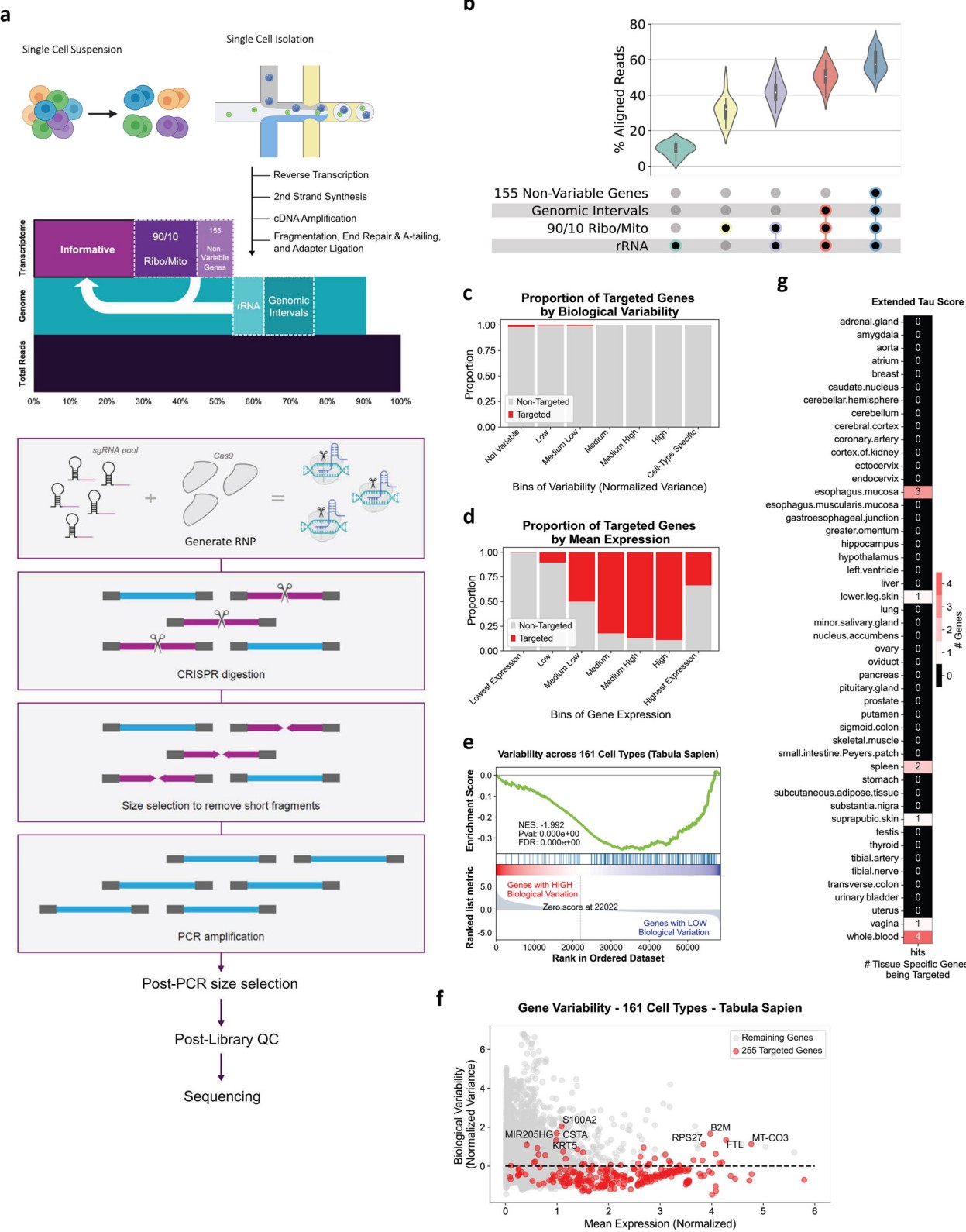

## scCLEAN increases sensitivity regardless of sequencing depth

First, single-guide RNA (sgRNA) arrays were designed against the genomic-defined intervals, rRNAs, and exonic regions of the 255 protein-coding genes, respectively, using an Ensembl GRCh38 human reference (see "Methods"). PBMCs were then isolated and used to generate full-length cDNA and subject to either processing with the standard 10 × 3' v3.1 method (10x-V3) (10,000 cells; $n$ = 1 channel) or

scCLEAN (scCLEAN) (30,000 cells; $n$ = 3 channels) each at a targeted sequencing depth of 25,000 reads per cell (see "Methods"). We first observed a 2-fold increase in non-targeted transcriptomic reads (hereinafter referred to as the informative transcriptome) with scCLEAN (Fig. 2a). We also highlight that the scCLEAN-targeted regions contributed to ~50% of reads, validating our in silico analysis of relative read contribution. We also sequenced the same 10x-V3 processed

**Fig. 1 | Evaluation of scCLEAN performance on human tissues and cell types.**
**a** Schematic representing the scCLEAN-mediated removal of abundant sequences from scRNAseq libraries. A single sgRNA pool was constructed from four unique sgRNA pools: (1) genomic intervals (teal), (2) non-polyadenylated rRNA (seafoam), (3) 90 ribosomal nuclear-encoded protein-coding and 10 mitochondrial genes (Ribo/Mito; purple), and (4) 155 non-variable genes (NVG; light purple).
**b** Distribution of the percentage of reads aligning to targeted regions after iteratively filtering reads corresponding to each of the four sgRNA guide-sets across 14 datasets. The median percentage breakdown is as follows: rRNA = 10%, Ribo/Mito = 34%, Genomic Intervals = 9%, and NVG = 5% for a cumulative sum of 58%.
**c–e** Analysis of the 255 gene panel using the Tabula Sapiens Consortium corresponding to 161 unique cell types across 24 organs. **c** Proportion of 255 targeted genes (red) from the total transcriptome across seven bins ranked according to (normalized variance) ranging from "Not Variable" to "Cell-Type Specific". Each bin contains an equal interval length between the minimum and maximum variances. The sum of genes in each bin were tallied and proportions of the 255 genes in each bin were evaluated. **d** Same as in c, except genes were ranked by mean gene expression (log(x + 1) normalized) and binned according to normalized mean gene expression ranging from "Lowest Expression" to "Highest Expression." **e** GSEA of the 255 gene panel within the Tabula Sapiens dataset between all 161 cell-types. Genes were ranked by normalized variance with the highest variance genes ranked at the top of the list. The normalized enrichment score (NES) is shown along with the $p$ value and false discovery rate. The location of the 255 targeted genes in the ranked list are indicated as blue dashes. **f** Gene scatter plot of biological variability (normalized variance) versus mean expression (log(x + 1) normalized). Dotted black line indicates zero variance. The top 8 genes within the 255 targeted gene panel are annotated with variance > 1. **g** Table summarizing the counts of tissue-specific genes from the 255 gene panel obtained from GTEx. Tissue-specific genes were quantified using the extended tau score metric and the intersection with the 255 gene panel were counted per tissue. Box plots depict the minimum, first quartile (Q1), median, third quartile (Q3), and maximum values for each group. Whiskers extend to the minimum and maximum values within 1.5×IQR from the quartiles. Detailed statistical values are available in Supplementary Data 3. Created in BioRender. Pandey, A. (2025) https://BioRender.com/g46i293.

sample to ~80,000 reads per cell and integrated a well-characterized PBMC dataset from scArches for direct comparison[32]. Additionally, we performed an in silico scCLEAN depletion (see "Methods") to compare the experimental depletion results.

Interestingly, when sorted by relative abundance of UMIs, scCLEAN reduced the fraction of UMIs contributed by genes with high abundance, reallocating UMI counts to molecules with relatively lower abundance (Fig. 2b). For instance, the top 200 ranked genes by total molecules across all cells contributed to 33% of median counts in the experimental scCLEAN conditions, 28% in the in silico condition, and 58% in the experimental control 10x-V3 condition. Perhaps the most relevant finding is the lack of increased complexity when oversequencing PBMCs to 80,000 reads per cell. Interestingly, the top 200 ranked genes in the deeply sequenced sample contributed to 61% of total UMIs, indicating that scCLEAN is necessary to improve coverage over lower abundance genes regardless of sequencing depth. This is further apparent when comparing experimental conditions only. In the control 10x-V3 experimental condition, non-targeted genes (36,346) accounted for 39% of UMIs per cell, while scCLEAN more than doubled (2.4-fold) the fraction of UMIs per cell to 92% which cannot be achieved with deeper sequencing (Fig. 2b, Supplementary Fig. 5). Providing further evidence, we next compared the library complexity which is defined as the fraction of total unique genes per total UMIs for each cell. On average, for every 1000 UMIs that were counted per cell, scCLEAN treated samples recovered a median of 650 unique genes resulting in a >2-fold boost in complexity when compared to control samples (Fig. 2c). Taken together, this further serves as evidence that the 255 genes primed for removal via scCLEAN contribute to a large fraction of UMIs in single-cell libraries. Furthermore, the enhanced biological resolution from scCLEAN cannot be recapitulated by sequencing more.

To confirm that scCLEAN-mediated reallocation directly translates to a greater separation of biological signal from noise, random matrix theory was employed to identify the optimized number of principal components. This method distinguishes between biologically relevant components and those associated with noise of a random matrix. scCLEAN increased the number of components included in secondary analysis from 7 in the 10x-V3 and deeply sequenced conditions to 10 (Fig. 2d). Given that principal component analysis (PCA) is often used to extract biological heterogeneity for dimensionality reduction, optimally selecting for more PCs is indicative of greater relied variance, and hence greater observed biological variation between populations of cells. Collectively, these data illustrate that the enhanced biological resolution from scCLEAN cannot be recapitulated by deeper sequencing.

After confirming successful removal of scCLEAN-targeted regions and subsequent reallocation to the remaining transcriptome for greater biological signal, we next sought to characterize any potential bias by performing Cas9-mediated cutting in vitro. Given that off-target effects of any CRISPR/Cas9 method both in-vivo and in-vitro are well documented, we aimed to assess scCLEAN's efficacy in downstream data interpretation[33,34]. We first generated a linear regression via pseudobulking UMI counts between 10X-V3 and scCLEAN and observed a high linear correlation coefficient ($r^2 = 0.96$–$0.97$) indicating minimal global bias (Supplementary Fig. 6). Although the general patterns were preserved, specific gene comparisons from pseudobulk analysis highlighted outliers from the linear regression indicating potential off-target cutting from CRISPR/Cas9. To rule out the possibility that differences were not due to stochasticity or sampling variance, in silico removal was utilized to identify all genetic regions that were targeted with scCLEAN sgRNAs (see "Methods"). Of the entire transcriptome, 15 additional genes showed decreased counts in the in silico scCLEAN model relative to 10x-V3 (Log2FC < −0.1), indicating transcriptomic coordinate overlap to scCLEAN-targeted genes. In the experimental scCLEAN condition, nine genes had fewer counts relative to 10x-V3 (Log2FC < −0.5), and of those genes, MTRNR2L12, S100A8, and EIF3F were not found to have overlapping transcriptomic coordinates from the in silico model, indicating off-targets resulting from sequence similarity to the sgRNA library. For the 9 genes with reduced counts, 8 were shared across three technical replicates indicating that the unexpected cutting from scCLEAN is minimal across samples and can thus be removed from downstream analysis (Supplementary Fig. 6).

## scCLEAN improves immune cell type identification and functional characterization

After applying multiple orthogonal methods to showcase an increase in the signal to noise ratio, we next sought to assess the performance of scCLEAN on cell clustering and subsequent cell-type identification. To do so, we proceeded with standard data preprocessing, normalization, PCA, k-nearest neighbor clustering, and finally dimensionality reduction for uniform manifold approximation and projection (UMAP) visualization (see "Methods"). For unsupervised reference-to-query label transfer, we leveraged a PBMC (Azimuth) reference atlas, consisting of ~500k cells. scCLEAN resolved 5 additional cell types: B intermediate, CD4 Proliferating, Erythrocyte, cDC1, and dnT[35]. Interestingly, one cell type was discordant between the two conditions, CD4 TEM in the 10x-V3 condition and CD4 CTL in the scCLEAN condition (Fig. 2e). In the 10x-V3 control, 32% of cells in the cluster were labeled as CD4 TEM, whereas with scCLEAN, 77% of cells in the cluster were labeled as CD4 CTL, indicating >2-fold higher confidence in the correct annotation of this cell type with scCLEAN (Supplementary Fig. 8). Lastly, we leveraged an orthogonal unsupervised deep learning method, which utilizes a deep embedding algorithm that iteratively

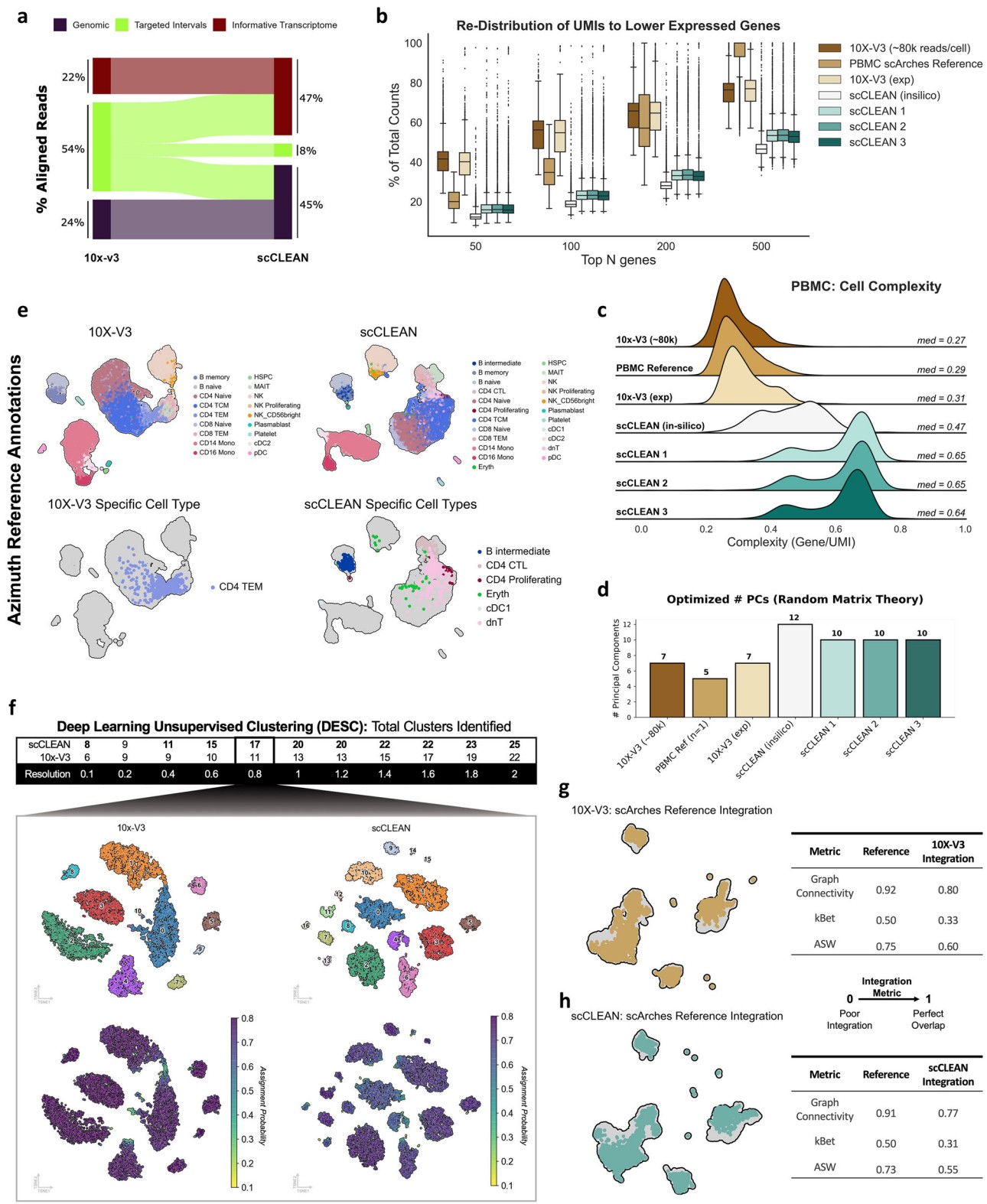

optimizes a clustering objective function, thus identifying the proper number of clusters as opposed to cell types[36]. Given the full spectrum of the user defined resolution parameter, scCLEAN consistently identified between 2 and 7 additional clusters while maintaining high confidence in cluster assignment (Fig. 2f). Multiple independent methods for cell clustering and annotation support scCLEAN's utility for the characterization of heterogeneous biology and cell-type identification.

In addition to cell label transfer, it is becoming a larger effort to integrate multiple experimental batched samples within the same latent space of consortium atlases[37]. Since scCLEAN shifts the library composition to lesser abundant genes, we utilized transfer learning framework scArches to confirm scCLEAN libraries can still be properly embedded into the latent space of a previously compiled PBMC reference atlas[32]. Because scArches uses a neural network model trained on single-cell counts data from biologically relevant genes

**Fig. 2 | scCLEAN increases cell genetic complexity enhancing PBMC cell type characterization. a** Sankey diagram illustrating the redistribution of the proportion of aligned reads from the standard 10x-V3 workflow (*left*) to the workflow incorporating scCLEAN (right) separated into three buckets. The 'Genomic' bucket (purple) represents non-targeted intergenic reads; "Targeted Intervals" bucket (green) represents reads from scCLEAN-targeted molecules; "Informative Transcriptome" bucket (red) represents reads aligned to the transcriptome excluding targeted molecules. Percentages of each bucket are shown. **b** Box plots displaying the distribution of UMIs (proportion of total) corresponding to the top 50, 100, 200, 500 expressed genes. Comparisons are between a PBMC sample sequenced to ~80,000 reads per cell (>3-fold deeper), an experimental 10x-V3 sample from the same batch, a PBMC reference atlas compiled from 3 publications using scArches, an in silico modeled scCLEAN condition assuming 100% read removal, and 3 technical experimental replicates of scCLEAN. **c** Ridge plots comparing the library complexity measured as the ratio of median genes to median UMIs per cell.

**d** Comparative bar plots displaying the optimized number of principal components calculated via random matrix theory to represent the biological signal identified. One scArches dataset was selected to ensure an accurate comparison between samples representing a single batch. **e** UMAP plots illustrating the cell types detected from query-reference mapping using the Azimuth PBMC dataset; 10x-V3 (*left*) with scCLEAN (*right*). 18 cell types with 1 uniquely identified within 10x-V3. 23 cell types with 6 uniquely identified within scCLEAN. **f** t-SNE clustering output derived from an unsupervised deep learning algorithm (DESC) iteratively spanning 11 different louvain resolutions starting with 0.1 and ending with 2 using 0.1 intervals. Representative t-SNE (*top*) clustering plots from a chosen resolution (0.8) showing 11 clusters in 10x-V3 and 17 clusters in scCLEAN with assignment probabilities shown below. **g, h** Metrics for integration accuracy from query-reference latent space projection using scArches. **g** UMAP plot for 10x-V3 (*left*) integration. Table for 10x-V3 (*right*) displaying integration metrics (graph connectivity, kBet, ASW). **h** Same as in (**g**), except showing metrics for scCLEAN.

---

only, we hypothesized that scCLEAN-generated counts data would efficiently integrate due to a higher observed signal of biologically relevant genes. Additionally, we leveraged the treeArches framework via scArches to generate cell type hierarchy trees to assess label transfer performance in addition to latent space integration (Supplementary Fig. 9) As expected, we observed no difference in the mapping efficiency into the latent space, and a negligible decrease in the integration performance according to 3 separate metrics (kBET, ASW, graph connectivity) (Fig. 2g, h)[38,39]. We underscore scCLEAN provides additional cell type resolution with no prohibitive effect on reference atlas integration.

### scCLEAN enhances single-cell transcript isoform quantification within PBMCs

To illustrate scCLEAN's applicability to third-generation sequencing, we applied the scCLEAN method to Pacific Biosciences' (PacBio) MAS-seq, a high-throughput single-cell library preparation method from droplet-generated cDNA (10x-V3)[40]. For MAS-seq analysis, we generated three technical sequencing replicates for both the 10x-V3 and scCLEAN conditions from 10,000 cells obtained from a single healthy donor. While in the second-generation sequencing application, scCLEAN is employed after fragmentation and Illumina adapter ligation, scCLEAN is applied to full-length cDNA prior to entering the MAS-Seq library preparation, further exemplifying the versatility of scCLEAN to various dsDNA intermediates (see "Methods"). A single pool of full-length cDNA was split and carried through with MAS-seq method alone or in combination with scCLEAN, thus enabling a direct cell-to-cell comparison of scCLEAN performance on the cell-barcoded cDNA (Fig. 3a). Comparable to the short-read results (Fig. 2a), scCLEAN reallocated reads to the non-targeted informative transcriptome resulting in an ~2.5-fold increase (Fig. 3b). When filtering for non-targeted genes only, the additional coverage led to a significant increase in transcript quantification associated with incomplete-splice-match (ISM), novel-in-catalog (NIC), and novel-not-in-catalog (NNIC) outputs from SQANTI3 (Fig. 3c). We then collapsed the transcript-level counts into gene-level counts to generate a gene x cell matrix for downstream analysis. After generating matrices, we evaluated the distribution of UMI counts associated with non-targeted genes and found that scCLEAN significantly boosted the median UMIs and median genes detected per-cell (Fig. 3d and Supplementary Data 4). These improvements were a direct result of reallocating UMIs from scCLEAN-targeted genes toward less abundant transcripts (Supplementary Fig. 10). We next sought to evaluate the collapsed gene-level information obtained from third-generation sequencing in clustering performance when comparing scCLEAN to the experimental control. Again, leveraging unsupervised label transfer using the same Azimuth PBMC reference as before, three additional cell types were annotated in the scCLEAN condition: cDC1, NK proliferating, and CD4 proliferating when compared to the control (Fig. 3e)[35]. Interestingly, in

both the second and third-generation sequencing applications, cDC1 and CD4 proliferating cells were only identified with scCLEAN. There was one cell type whose annotation was incompatible between conditions, labeled as CD8 TEM with 10x-V3 and CD4 CTL with scCLEAN (Fig. 3e). For both labels, the confidence in the annotation was low (<40%), indicating a heterogeneous cluster causing a switch in the classification (Supplementary Fig. 11). Because we collapsed the transcript-level counts to the gene-level for dimensionality reduction, we focused on elucidating transcript-level isoform coverage of marker genes leading to the successful classification of rarer cell types. Specifically for the cell type annotated cDC1, we highlight BTLA and ENPP1 which are known marker genes for cDC1[41,42] (Fig. 3f). For both BTLA and ENPP1, scCLEAN facilitated full-length exon coverage (read depth > 2) without necessitating deeper sequencing. Importantly, for these marker genes, the MAS-Seq method alone was unable to generate meaningful exon coverage. We would also like to highlight the increase in read depth coverage for COX5A, a nuclear-encoded mitochondrial electron transport chain subunit, which showcases scCLEAN's specificity for mitochondrially-encoded transcripts while leaving the nuclear-encoded mitochondrial counterparts unperturbed. The benefit of reallocating reads to less abundant molecules in third-generation sequencing applications underscores scCLEAN's ability to improve both high-level gene counts detection and lower-level exon coverage of biologically relevant transcripts.

### scCLEAN differentiates phenotypic states in cardiovascular smooth muscle cells

While scCLEAN greatly improves the analysis of PBMCs which are a well-characterized heterogeneous mixed population of cells, we next applied the method to a homogenous cell population notorious for transitioning phenotypically between many functional states[43–47]. VSMCs are regularly studied to understand the pathogenesis of cardiovascular disease; but it is unclear to the extent these cell types exhibit phenotypic plasticity under various environmental conditions, especially when compared between coronary and pulmonary origins. Given an individual cell type population was isolated, we transitioned our focus from cell type classification of heterogeneous samples to other methods of parsing biological processes. We hypothesized that VSMCs isolated from coronary and pulmonary arteries represent distinct cell state lineages that can be traced to understand potentially underlying biological processes that control disease progression as it pertains to cardiovascular disease[48]. Furthermore, we hypothesized that scCLEAN would illuminate previously obscured biological processes given its ability to garner more insight from less abundant transcripts.

VSMCs from four matched healthy donors isolated from both coronary and pulmonary arteries, representing 49% and 51% of the total cell population, respectively, underwent 10×3' v3.1 droplet generation to create 16 cDNA pools (Fig. 4a). Again, as previously done

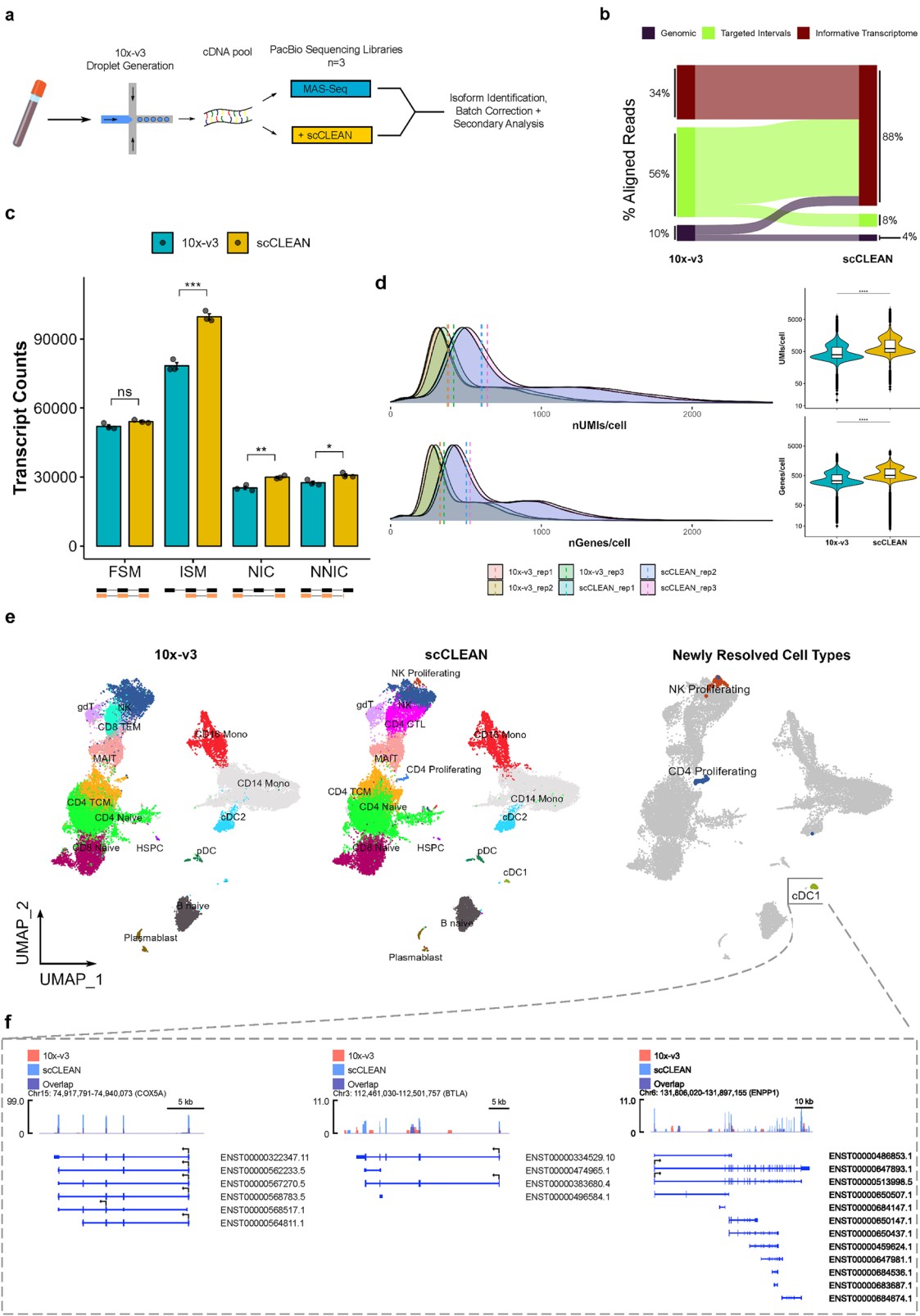

with the PBMCs, each cDNA pool was split and processed with the standard v3.1 method alone (10x-V3) or with the addition of scCLEAN and sequenced on an Illumina instrument. Initial scCLEAN performance metrics compared similarly to those observed in PBMCs, successfully reallocating 35% of aligned reads which increased the number of informative transcriptomic reads by 58% and increased the proportion of informative UMIs from 66% to 95% (average across 8 samples) (Supplementary Fig. 12).

We then proceeded with data QC, dimensionality reduction, and trajectory inference. Trajectory inference has become a trademark in single-cell data analysis due to its ability to extract dynamics from a snapshot of data lacking intricate and laborious time-courses[49]. To examine such dynamics, we employed CellRank, a trajectory inference tool designed to extract cell state dynamics, to investigate the phenotypic manifold of potentially transitioning VSMCs[50]. When comparing coronary and pulmonary VSMCs, the density of cells remained

**Fig. 3 | scCLEAN improves immune cell type characterization with single-cell MAS-Seq. a** Schematic illustrating experimental workflow to incorporate scCLEAN into the MAS-Seq method. An amplified barcoded-cDNA pool generated with the 10X Genomics Chromium Controller was split and processed with MAS-Seq alone or in combination with scCLEAN (**b**) Sankey diagram as in Fig. 2a, except showing the flow from the MAS-seq method alone (*left*) to MAS-Seq with scCLEAN (*right*). **c** Transcripts aligning to the 255 gene panel were masked and the subsequent total number of identified transcript counts were binned into four categories via SQANTI3: full splice match (FSM), incomplete splice match (ISM), novel-in-catalog (NIC), and novel-not-in-catalog (NNIC). Differences in transcript quantities between 10x-V3 (blue) and scCLEAN (yellow) for each category were measured using a two-tailed *t*-test (*n* = 3; ns = *p* > 0.05, * = *p* ≤ 0.05, ** = *p* ≤ 0.01, *** = *p* ≤ 0.001). Error bars represent ±SEM. **d** Transcripts aligning to the 255 gene panel were masked and (*left*) histograms of the shift in distributions of UMIs per cell and genes per cell

comparing 10x-V3 and scCLEAN are shown. (*right*) Violin plots of UMIs per cell and genes per cell. Each replicate (*n* = 3) is shown. Significance was quantified using a two-tailed Wilcoxon rank-sum test with Bonferroni correction (**** = *p* ≤ 0.0001). **e** UMAP plots representing the query-to-reference labels using the Azimuth PBMC reference. Comparison between 10x-V3 (*left*), scCLEAN (*middle*), and the cell types uniquely identified within scCLEAN (*right*). **f** Cells labeled as cDC1 within the scCLEAN condition were selected and treated as bulk. Track plots indicate read coverage over top 3 cDC1 markers (*COX5A*, *BTLA*, and *ENPP1*) comparing 10x-V3 (red), scCLEAN (blue), and overlap (purple). Ensembl transcript IDs are annotated below. Box plots depict the minimum, first quartile (Q1), median, third quartile (Q3), and maximum values for each group. Whiskers extend to the minimum and maximum values within 1.5 × IQR from the quartiles. Detailed statistical values are available in Supplementary Data 4.

comparably distributed across both scCLEAN and 10X-V3 throughout the trajectory indicating undetected bias in global cellular distribution along the projected data (Fig. 4b). After confirming successful data integration, we then constructed pseudo-temporal trajectories utilizing the cytoTRACE computational framework to infer differentiating states associated with each VSMC arterial origin (Fig. 4c)[51,52]. After computing terminal macrostates using Markov chains[53], one lineage was identified in the 10x-V3 condition, while two lineages were identified with scCLEAN (Fig. 4d). According to the individual cell-fate potentials, 64% and 36% of cells in the scCLEAN condition were associated with lineage 1 and 2, respectively (absorption probability > 0.5) (Fig. 4e). Excitingly, scCLEAN identified an additional terminal state of VSMCs consistent with the number of lineages present in the data.

However, we next sought to resolve if the arterial origin had a direct concordance to each lineage state transition. Because the Markov-chain simulates random walks for each cell towards a terminal state, each cell is thus assigned a probability of transitioning toward a final state. Using this information, we generated a binary classification system to relate the absorption probabilities of each terminal state to arterial origin. After comparing artery identity with lineage absorption (Fig. 4f, g), it was evident that each terminal state was representative of the artery of origin. Receiver operating characteristic analysis confirmed the classification of lineage to artery identity with an AUC of 0.99 (Fig. 4h). To validate this interesting finding and ensure the robust classification was inherent to scCLEAN, two terminal lineages were manually computed in the 10x-V3 condition, but discriminatory power was non-existent (AUC = 0.51) (Supplementary Fig. 13). As a result, scCLEAN enabled blind algorithmic classification of VSMC lineage to artery identity with outstanding accuracy, in a system that was otherwise indistinguishable.

### scCLEAN identifies an inflammatory network unique to coronary VSMCs

To expand on the clear identification of distinct lineages corresponding to coronary and pulmonary arteries, we next examined driver genes responsible for the trajectory-specific dynamics. Interestingly, scCLEAN enabled the identification of coronary and pulmonary specific driver genes explaining cellular fate dynamics (Fig. 5a). To further elucidate, we generated a principal tree with discrete branches and bifurcations (Fig. 5b)[54]. By doing so, this enabled the distinction between lineage markers, genes whose pattern of expression across pseudotime are unique to either the coronary or pulmonary terminal state (Fig. 5c–e), and transition markers, genes responsible for initiating the separation (Fig. 5f). The top coronary transition markers (IL1A, IL1B, IL11, IL32, CCL2, CXCL5) are key players in inflammation and are distinctly associated with coronary VSMCs (Fig. 5f, g). Gene-set enrichment analysis (GSEA) further confirmed the coronary specific inflammatory signatures, with considerable overlap with inflammatory response and TNF-alpha signaling via NF-kB (Fig. 5h). Next, we identified regulatory mechanisms associated with

each coronary and pulmonary lineage to inform future exploration and potential avenues for therapeutic intervention. We found that nuclear factor erythroid 2-related factor 2 (NFE2L2) was activated in 91% of cells in the pulmonary lineage but only 1% of cells in the coronary lineage (Fig. 5i). NFE2L2 is pivotal in the regulation of antioxidant proteins that protect from injury and inflammatory induced oxidative damage[55]. Considering oxidative stress contributes to the development of vascular diseases such as atherosclerosis, the lack of NFE2L2 regulon activation in coronary cells may provide insight into early disease onset and warrants further investigation.

## Discussion

Herein we describe scCLEAN, a molecular add-on to improve the biological resolution of scRNAseq datasets that suffer from poor characterization of lowly expressed transcripts. Instead of focusing on higher total cellular throughput, scCLEAN represents an unconventional approach to enhance sensitivity by globally recomposing the library to enhance complexity of lesser abundant molecules. In PBMCs, scCLEAN demonstrates an improvement in immune cell-type identification and functional characterization from both second-generation and third-generation sequencing technologies. Similarly, in VSMCs, scCLEAN revealed nuanced but important differences in cell identities from different vascular beds, which was not possible by standard scRNAseq analysis. By redistributing ~50% of reads in-vitro, scCLEAN increases the biological variance incorporated into the latent representation according to multiple orthogonal methods of dimensionality reduction, an improvement that cannot be recapitulated by deeper sequencing.

Further analysis using scCLEAN enabled the delineation between nuanced transitioning states within a single homogenous cell type. scCLEAN uniquely characterized pre-symptomatic, basal level differences between coronary VSMCs and pulmonary VSMCs, establishing a stark juxtaposition of similar but transcriptionally transitioning vascular beds. Coronary artery specific inflammation, as validated by the top coronary lineage markers as well as GSEA, is considered a hallmark process of atherogenesis[56]. Thus, scCLEAN facilitated a transcriptional distinction between coronary and pulmonary VSMC plasticity in non-symptomatic donors, providing insight into the underlying coronary predisposition to cardiovascular disease.

While we demonstrate the increase in resolution for difficult to distinguish biological signals, we also emphasize that a panel of 255 protein-coding genes were removed from the analysis. While we resolved additional cell types and discovered unique gene signatures using this method, we provide a relative cost-benefit analysis of scCLEAN via projection onto GTEx tissues and the Tabula Sapiens consortium, further establishing a comprehensive guide for researchers to determine the applicability of scCLEAN on their sample[27]. Additionally, scCLEAN activity was not predicted within all diseased conditions and application should be decided according to the sample specific variance of the 255 genes. Rather, the optimal use

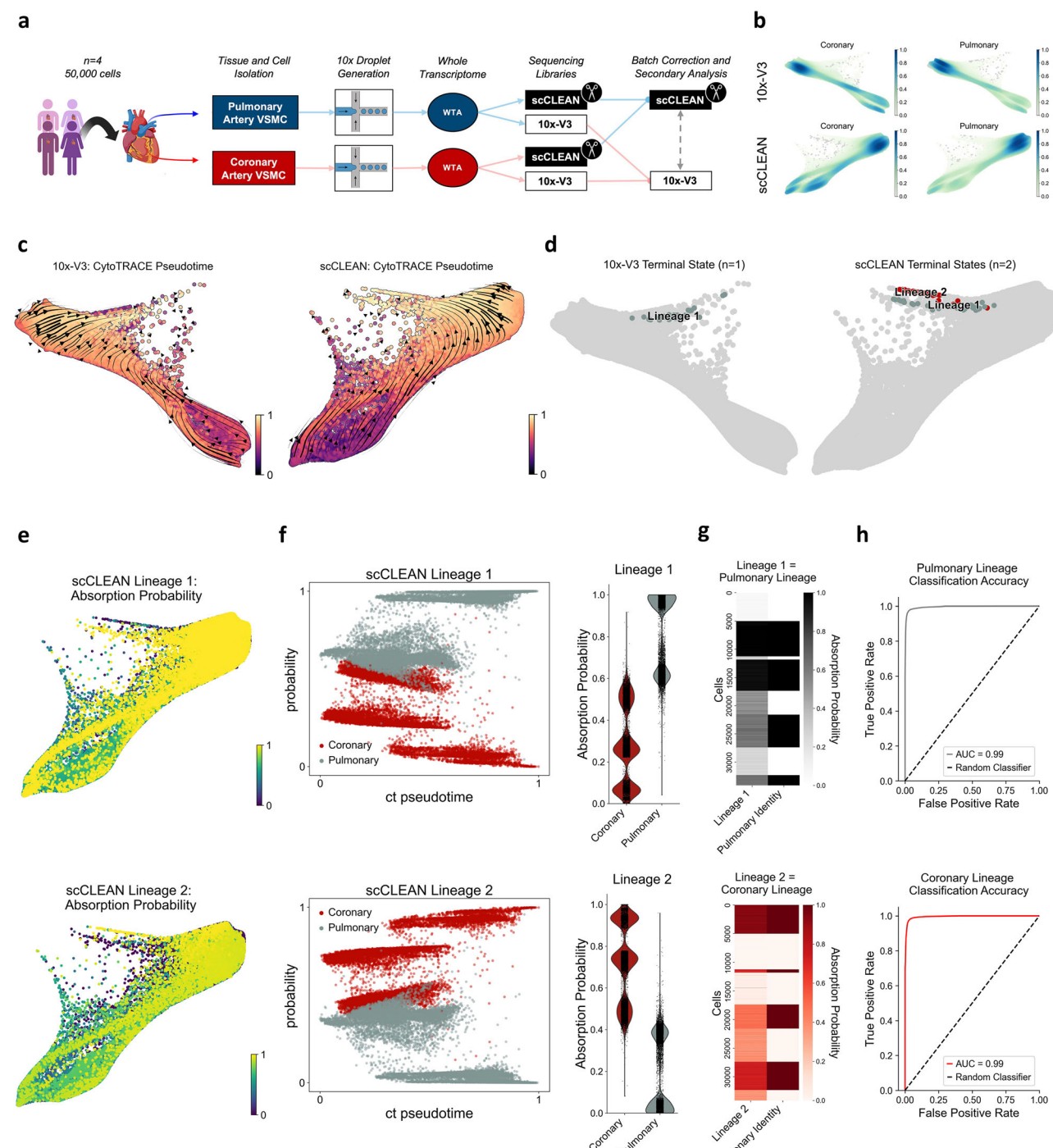

**Fig. 4 | scCLEAN identifies 2 lineages of VSMCs corresponding with the artery of origin. a** Schematic figure of experimental workflow. A total of ~50,000 VSMCs were isolated from 4 donors and 2 tissue locations: coronary and pulmonary artery. **b** Normalized density of cells along the force directed layout (FLE) according to their artery of origin comparing 10x-V3 (*top*) with scCLEAN (*bottom*). **c** RNA velocity stream plots along pseudotime (CytoTRACE) mapped onto the FLE embedding. All detected genes are utilized to automatically calculate early pseudotime (black) to late pseudotime (yellow) according to the number of genes expressed. Black arrows indicate direction of transcriptional transition. **d** The total number of terminal states identified using CellRank comparing 10x-V3 (*left*) and scCLEAN (*right*). Number of lineages are shown. Optimal terminal states were found using Schur decomposition (gap in the real portion of eigenvalues) and then refined according to stationary distance of the coarse-grained Markov transition matrix (non-zero distance). **e** Only scCLEAN is shown (>1 lineage). Absorption probability

of each cell belonging to each lineage and thus differentiating along pseudotime into either of the 2 terminal states, corresponding with lineage 1 (*top*) and lineage 2 (*bottom*). Yellow represents a 100% probability of that cell belonging to that lineage. **f** (left) Lineage absorption probability (lineage 1 = top, lineage 2 = bottom) plotted as a function of each cell's position along the differentiation trajectory (ct pseudotime). Colors reflect tissue origin: coronary (red) or pulmonary (gray) artery. (right) Violin plots of absorption probabilities across coronary and pulmonary arteries for each lineage. **g** Heat matrix illustrating the relationship between the quantified absorption probability of each cell belonging to each lineage compared to the ground truth arterial identity of that cell. **h** Receiver operating characteristic (ROC) depicting the classification performance of identifying each artery to each lineage (both pulmonary and coronary lineage AUC = 0.99). Created in BioRender. Pandey, A. (2025) https://BioRender.com/r94y754.

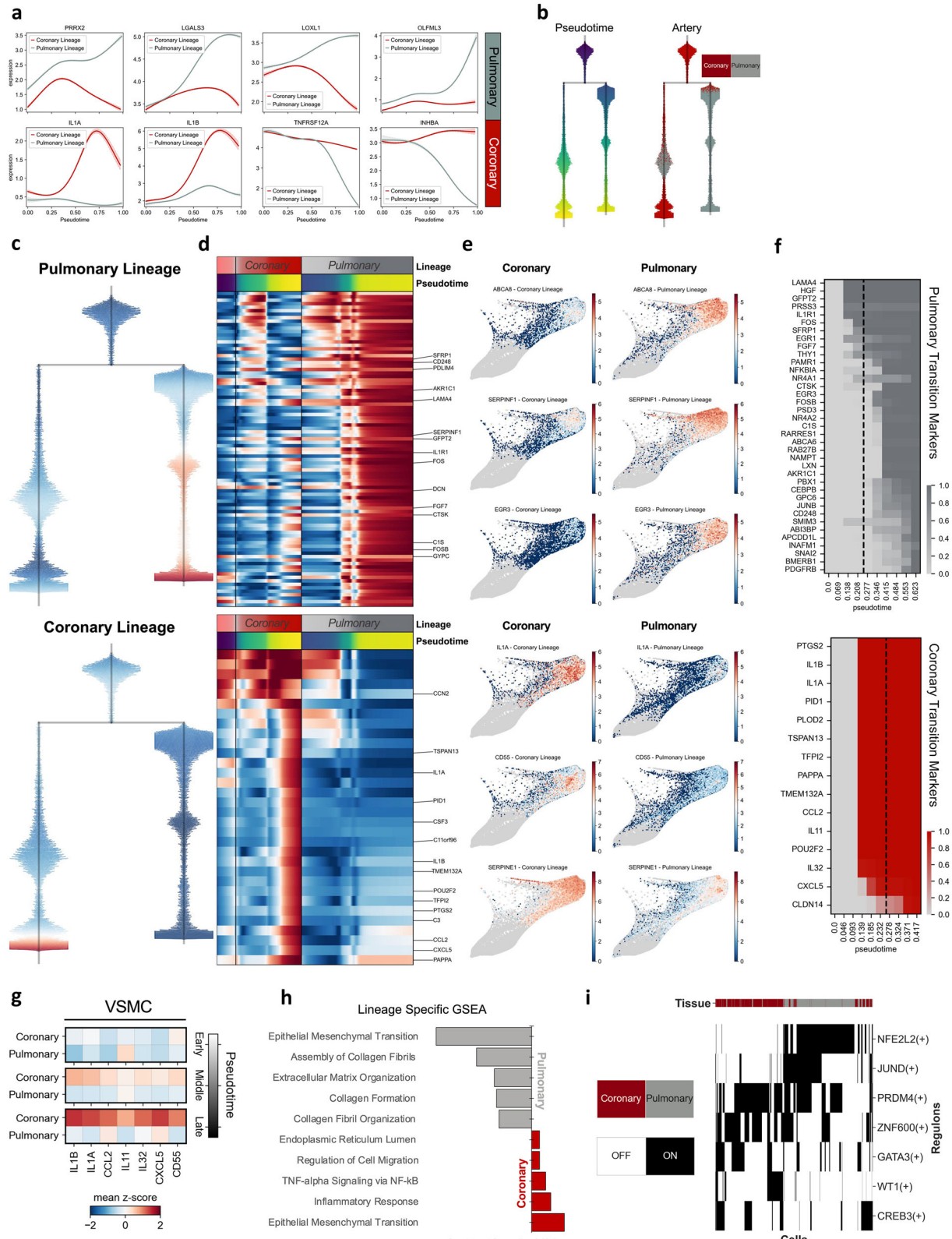

case for scCLEAN is when users are interested in either increasing the sensitivity of detection within their scRNAseq library or interpreting complex polygenic regulatory networks and cell type/state signatures that are difficult to parse or that have otherwise been obscured by the presence of abundantly sequenced molecules.

Overall, scCLEAN holds substantial potential to improve single-cell methodologies beyond short and long-read 3' scRNAseq assays.

We emphasize that scCLEAN was tailored to remove uninformative molecules curated against polyA-captured scRNAseq libraries generated with the 10X Genomics v3.1 3' method. Future work would focus on designing optimized guide arrays for other single-cell sequencing data such as spatial transcriptomics and Perturb-seq. Furthermore, we reason that as the field continues to progress toward multi-omics, combining multiple modalities such as the genome, transcriptome,

**Fig. 5 | Coronary artery lineage of VSMCs uniquely expresses an inflammatory network. a** Genes with the highest correlation associated with each terminal state. Average gene expression sampled from 200 cells with an absorption probability >0.5 associated with each lineage (pulmonary lineage = top, coronary lineage = bottom) and mapped along pseudotime. **b** Dendrograms representing the principal graph (scFATES) inferred from scCLEAN terminal states. (*left*) Blue represents early pseudotime and yellow represents late pseudotime. (*right*) Cells colored according to arterial origin (coronary = red, pulmonary = gray) projected along pseudotime. **c** Tree diagram highlighting onset of characteristic gene signatures (red) associated with pulmonary lineage (*top*) and coronary lineage (*bottom*). **d** Gene expression heatmaps of highest specificity (effect > 0.3) depicting early root cells (*left*), coronary lineage (*middle*), and pulmonary lineage (*right*). The smoothed gene expression is shown along pseudotime and colored by expression (low = blue and high = red). Gene specific pseudo-temporal activation to the pulmonary lineage

(*top*; n = 91) and to the coronary lineage (*bottom*, n = 32). The top 15 genes for each arterial origin are depicted. **e** Feature plots of log(x + 1) normalized expression on the FLE embedding comparing the expression of 3 representative lineage markers specific to the pulmonary (*top*) and coronary (*bottom*) lineages. **f** Gene transition heatmap of top markers for the pulmonary (*top*) and coronary (*bottom*) bifurcation to reflect early decision-making processes prior to separation. The matrix contains inclusion timing for each gene in each probabilistic lineage projection. **g** Z-scored mean expression of inflammatory genes comparing coronary and pulmonary across three equivalent bins of pseudotime (early, middle, late). **h** The top five enriched lineage-specific pathways found using GSEA with all lineage-specific genes (correlation greater than 0.3). The combined score (y-axis) incorporates gene overlap and statistical significance. **i** Top regulons with differential expression between coronary and pulmonary lineages. Binarized regulon activity (SCENIC).

epigenome, and proteome from a single cell will only exacerbate the noise contributed by each technique. scCLEAN provides an opportunity to address these bottlenecks and enhance biological resolution.

# Methods

## Ethics statement

Research for this manuscript complies with all relevant ethical regulations and requirements. Human donor samples were de-identified for the entirety of the study, with no identifiable data given to study investigators. All studies of the human cells (PBMCs and VSMCs) were approved by the Institutional Review Board of Scripps Clinic/Scripps Research Institute (IRB-18-7266) and Tulane University (2022-1434-TUHSC). Written informed consent from all donors was obtained as per the original study IRB and subsequently approved by the above study IRBs.

## sgRNA design and generation

We curated a set of 14 different scRNAseq datasets from Sequence Read Archive (SRA) for various human sample types. Note that samples in each of these datasets were prepared in accordance with the 3' 10x Chromium V3 technology (10X Genomics, Pleasanton, CA). The human genome (GRCh38) was used for all sgRNA designs.

**sgRNA filtering and synthesis.** The Jumpcode Genomics' (San Diego, CA) proprietary sgRNA design pipeline was employed to design CRISPR sgRNAs. A 20mer target sequence adjacent to NGG (PAM) sites were selected as candidate targets. Off-target cleavage in exonic regions (for manufacturability) of all remaining annotated protein-coding genes was minimized by excluding matching sgRNA sequences with sequence homology up to 3 mismatches. The remaining sgRNAs were again filtered to remove high and low GC sequences, homopolymers <4 nt in length, and dinucleotide stretches. The final guide set was generated by selecting guides with high in-vitro cleavage prediction scores using CRISPick[57]. Finally, sgRNAs were filtered for maximum density with an inter-guide spacing adapted from Gu et al.[23] DNA oligonucleotide pools were synthesized in an array format and purchased from Agilent Technologies (Santa Clara, CA), containing the 5' bacteriophage T7 RNA polymerase promoter sequence, a target-specific 20mer sequence, and a 100mer trans-activating CRISPR RNA sequence. The DNA oligonucleotide pools were then amplified by PCR and converted to RNA by in-vitro transcription using the SureGuide gRNA Synthesis Kit purchased from Agilent Technologies following manufacturer's recommendations. The products of in-vitro transcription were then treated with DNase I, provided in the SureGuide gRNA Synthesis Kit, column purified, and pooled in equimolar ratios to form the final sgRNA pool.

**Ribosomal RNA (rRNA), mitochondrial protein-coding, and ribosomal protein-coding sgRNA design.** The rRNA sgRNAs were designed against the mitochondrial 12S and 16S genes and nuclear 5S, 5.8S, 18S,

and 28S rRNA genes derived from the 45S precursor rRNA transcript, resulting in 435 unique sgRNAs. Additionally, sgRNAs were designed against 10 mitochondrially-encoded protein-coding genes and 90 nuclear-encoded S and L ribosomal protein-coding genes resulting in 4195 unique sgRNAs.

**Genomic Intervals sgRNA design.** For each SRA dataset, the STAR aligner was used to map reads to reads were aligned to a reference index built from Ensembl transcriptome (GENCODE v32/Ensembl 98) passing the flag --twopassMode Basic to maximize the read alignment[58]. All reads that overlapped with protein-coding transcripts were removed. The filtered bam files from each dataset were then used to calculate the depth of coverage over 500 base-pair windows across each chromosome of the human genome with assessment that no transcriptomic regions were targeted. The 500 base-pair genomic intervals were then ranked according to average depth of coverage for each dataset. The genomic intervals with the highest shared coverage across all 14 datasets were selected followed by the combination of the top dataset specific genomic intervals using Jumpcode Genomics' proprietary pipeline. We then filtered for off-targets as described above.

**Non-variable genes (NVG) sgRNA design.** To identify a set of highly expressed protein coding genes that did not play a significant role in identifying cell types across a diverse set of human tissues, all 14 SRA datasets were aligned to the same transcriptome as described above using cellranger count and underwent standard secondary processing according to Pegasus (https://github.com/lilab-bcb/pegasus)[59]. For each dataset, the top 5000 highly variable genes were filtered from the data. All remaining genes were then ranked by mean expression for each dataset. We then employed Jumpcode Genomics' proprietary pipeline to curate the intersection of genes across all 14 datasets with the highest expression followed by sgRNA design and off-target filtering as described above. The results was 155 selected genes that were then assessed for identity consistent with primarily housekeeping genes using PanglaoDB[28].

**In-silico modeling of scCLEAN.** The underlying basis of the in silico modeling revolves around the hypothesis that when molecules are removed, they are randomly replaced by other molecules in the library according to each molecule's relative abundance. Considering scCLEAN is performed prior to amplification bias, UMIs associated with lowly expressed genes that would otherwise be outcompeted are recovered, fundamentally changing the composition of the library relative to the standard condition. Consequently, sequencing represents random sampling from two dissimilar pools of molecules, lending to the enhanced detection sensitivity from scCLEAN. To replicate this process in silico, this model is comprised of two simple, fundamental steps: replicate a scCLEAN library, and mimic random sampling from sequencing.

For the dataset representing the 10x-V3 condition, the raw reads were downloaded from the dataset labeled "5k PBMCs from a Healthy Donor (Next GEM)" (5k_pbmc_v3_nextgem_fastqs.tar). For each sample, all reads from each lane were concatenated. The scCLEAN condition was computationally simulated from the 10x-V3 condition.

First, we duplicated the 10x-V3 condition that we are going to simulate into scCLEAN. These duplicated reads were then aligned to a custom STAR index representing all regions targeted for removal by the sgRNA incorporated into the molecular kit. Specifically, the STAR index was built from a combination of fasta files: sequences representing the genomic but not transcriptomic 500 bp intervals, sequences representing the 255 protein coding genes obtained from Ensembl Biomart (Unspliced gene with 100 bp flanking region on each side). Alignment to this index simulates CRISPR-Cas9 incubation and assumes 100% efficiency in removal of targeted material. The STAR alignment command included the flag –twopassMode Basic and 46% of reads were removed. Next, to remove the non-polyadenylated ribosomal RNA, the sortmerna tool (https://github.com/biocore/sortmerna) was run with default parameters, referencing an index composed of fasta sequences from all rRNA compiled from NCBI[60]. A tool designed specifically to remove rRNA was used due to the poor alignment of rRNA using traditional RNA-seq aligners (4% of reads were removed).

Second, to mimic the effect of random sampling from two molecular pools of differing composition, the scCLEAN simulated reads were down sampled using seqtk to an estimated depth to recover roughly 25,000 reads per cell. The bash script repair.sh from BBmap repository (https://github.com/BioInfoTools/BBMap) was used to down sample the read 1 file while maintaining the paired barcode information. The 10x-V3 condition was down sampled to the same exact depth. To compare, samples were then aligned using cellranger count to the standard 10x transcriptome (refdata_gex_GRCh38_2020_A).

### PBMC samples

**Sample extraction.** Blood was obtained from the Scripps Normal Blood Donor Service from anonymized donors and fully compliant with approved IRB protocols for human subjects. Blood was collected and prepared for downstream single-cell applications. Blood was collected in BD Vacutainer cell preparation tubes (CPTs) with sodium citrate in 4 mL aliquots. Blood samples were processed within 2 h of collection. CPTs were centrifuged at 1700 rcf for 15 min at room temperature to separate the PBMC layer using polyester gel and density solution in CPTs. The top layer of liquid containing PBMCs was decanted into 50 mL conical tubes. 1X PBS was added to reach 50 mL total volume and tubes were centrifuged at 300 rcf for 10 min at room temperature. Supernatant was discarded and pellets were resuspended with 10 mL of 1X PBS. 1X PBS was added to reach 50 mL total volume and tubes were centrifuged again at 300 rcf for 10 min at room temperature. Supernatant was discarded and pellets were resuspended with 4 mL of 1X PBS + 0.04% BSA. Samples were then used for downstream single-cell applications.

**PBMC single-cell library generation and scCLEAN removal of abundant molecule.** The PBMC suspension was prepared for droplet generation according to the manufacturer's suggested protocol (10X Genomics, CG00053 Rev C). Briefly, four channels, with a targeted cell recovery of 10,000 cells were loaded on the 10x Chromium Controller. Using the Chromium Next GEM Single Cell 3′ Kit v3.1 (16 rxns PN-1000268), full-length barcoded cDNA was generated and 120 ng of cDNA from each channel was carried through step 3.4–Post Ligation Cleanup – SPRIselect. In parallel, one control ($n = 1$) library was generated to completion manufacturer's recommendations while three libraries were subjected to scCLEAN depletion ($n = 3$) after first eluting in 16 μL nuclease-free water; a slight modification in step 3.4. Immediately after post ligation cleanup, 15 μL of each library was incubated at room temperature with Cas9 pre-complexed with the sgRNA pool

described above using the CRISPRclean Single Cell RNA Boost Kit (Jumpcode Genomics) for 1 h at 37 °C. After incubation, a 0.6x bead-based size selection was performed using AMPure XP beads to remove the cleaved fragments and eluted in 30 μL nuclease-free water. The Chromium Next GEM Single Cell 3′ protocol v3.1 was proceeded from Step 3.5 onwards as outlined by manufacturer's recommendations. Each library was uniquely indexed using the Single Index Kit T Set A (10X Genomics, PN-1000213).

**Off target Identification.** With scCLEAN, the potential 58% of targeted reads are randomly re-assigned to other molecules in the library. This evenly boosts the abundance of non-targeted, informative transcripts. Consequently, differential expression methods such as DESeq2 cannot be used to accurately compare because DESeq2 identifies a universal change in library coverage as a change in sequencing depth and normalizes the reads accordingly[61]. DESeq2 models the counts as a negative binomial with a fitted mean, which is composed of a sample specific size factor. This sample specific size factor alters the counts of the scCLEAN condition, masking both the benefit as well as the off targets[8]. Consequently, a naïve approach to compare changes in gene expression should be used when identifying potential off targets. For example, we recommend down sampling the conditions to the same sequencing depth, running the analysis in parallel, treating the samples as bulk, and performing a direct comparison.

**Sequencing.** DNA concentrations were assessed using fluorometric quantification using the Qubit 4.0 Fluorometer (Thermo Fisher) and fragment sizes were analyzed using the Agilent BioAnalyzer 2100 (Agilent Technologies). Libraries were normalized to 2 pM, pooled, and diluted to achieve 600 pM loading concentration before loading onto a NextSeq 2000 P3 flow cell (using a NextSeq P3 v3 100 cycle kit). The following sequencing schematic: R1: 28 base-pairs, Index1: 8 base-pairs, R2: 91 base-pairs was employed. For subsequent downstream analysis, all samples were subsampled to 310 million reads (~27k reads/cell) using 'seqtk sample.'

### Dimensionality reduction

While dimensionality reduction is ubiquitous for single-cell data visualization, it is often performed using different techniques, such as the canonical principal component analysis (PCA)[62] combined with 2D-visualization[63–65], diffusion based[66–68] and deep-learning based methods[69–71]. We thus sought to characterize how scCLEAN affects the performance of each type of algorithm, in particular, the ability to retain biological variance within the reduced representation.

For PCA, MAGIC, and the autoencoder framework, molecular cross validation (MCV) (https://github.com/czbiohub/molecular-cross-validation) was used to select the optimized dimensionality reduction parameter for each strategy[72].

**PCA.** Initial filtering mimicked what was performed on PBMC data considering PCA is a necessary component in Seurat secondary analysis and UMAP visualization: cells removed in the top 1% of UMI counts, fewer than 500 UMIs, fewer than 200 genes, and genes removed that are expressed in fewer than 3 cells. Using Scanpy (1.9.1), the data was then further processed using the functions: normalize_total (target_sum = 1e4), log1p (default settings), and highly_variable_genes (default settings). The optimal number of principle components was then identified using the MCV fuction mcv_calibrate_pca with the arguments recipe_seurat and max_pcs = 60. The more principal components, the greater biological variance is retained within the latent representation.

**MAGIC.** Each Cell Ranger output filtered_feature_bc_matrix.h5 file was processed and pickled using the MCV python script process_h5ad.py with the following flags: –seed 2, –min_counts 500, –min_genes 200,

and −min_cells 3. The output pickle files then underwent a grid search and were calibrated using the MCV loss using the script magic_sweep.py, which uses the diffusion based, single-cell imputation tool MAGIC (Markov affinity-based graph imputation of cells)[73]. To reduce the search space and focus on diffusion t, the script was manually tweaked, changing the default for the −neighbors flag to (10,11), the default for the −components flag to (50,51), and the default for the −time flag to (1, 16). The optimized script was then run with the following flags: −seed 10, −data_split 0.9, −n_trials 10, −genes 2000. The optimal diffusion t was selected according to the lowest MCV loss calculated from 10 neighbors and 50 principal components.

For diffusion-based dimensionality reduction algorithms, the key parameter is how much to diffuse the data. Diffusion can be intuitively understood as data smoothing, the more the diffusion, the noise is smoothed over and effectively removed, but so is the biological heterogeneity. The less diffusion, and the original biological heterogeneity is retained, but so is all the associated noise. The fact that scCLEAN requires significantly less diffusion than 10x-V3 can be interpreted as a reduction in noise and a greater preservation of biological signal.

**PHATE.** PHATE (1.0.7) (https://github.com/KrishnaswamyLab/PHATE) was utilized in order to test the effect of scCLEAN on a network diffusion-based algorithm in addition to MAGIC as well as to ensure the results were not a by-product of the MCV optimization algorithm[74]. PHATE automatically calculates the proper diffusion component t internally using Von Neuman Entropy. To adhere as closely as possible to the guided tutorial, scprep (1.1.0) (https://github.com/KrishnaswamyLab/scprep) was used for initial data processing. Empty cells and genes were removed using the scprep functions filter_empty_cells and filter_empy_genes respectively. The libraries were then normalized and transformed using normalize.library_size_normalize and transform.sqrt. The phate.PHATE operator was then initialized with the arguments n_jobs = −2 and random_state = 10. The entropy was then calculated using the function _von_neumann_entropy with the argument t_max = 100 and the knee point was selected running vne.find_knee_point.

**Autoencoder.** Each cell ranger output filtered_feature_bc_matrix.h5 file was processed and pickled using the MCV python script process_h5ad.py with the following flags: −seed 2, −min_counts 500, −min_genes 200, −min_cells 3, −n_genes 5000, and −n_cells 10000. The data was additionally subset to 10,000 cells and 5000 genes to reduce run time. The MCV script autoencoder_sweep.py was altered to enable parallelization across 4 GPUs. The MCV autoencoder model was altered to enable parallelization using torch.nn.DataParallel. The optimized script was then run with the following flags: −seed 2, −gpu 1, −data-split 0.9, −learning_rate 0.1, −pois, −dropout 0.0, −layers 128, −max_bottleneck 7. The gpu flag identifies the index of one of the four potential GPUs to act as the main staging area for every input tensor to be provided as well as all final outputs to be gathered. The scCLEAN condition was ran with −gpu 0 and the 10x-V3 condition was ran with −gpu 1 for them to run simultaneously. The optimal bottleneck size was selected according to the minimum MCV loss.

Similar to the number of principle components in PCA, if the bottleneck layer is larger, then more biological variability is let through, but so is more noise. Considering the optimal width for the scCLEAN condition was greater than the 10x-V3 condition means more signal could be incorporated due to a reduction in noise. We highlight that training and the proper use of an autoencoder involves the optimization of many hyperparameters outside of the bottleneck width. Consequently, we tried to reduce the process to as simple a workflow as possible. We note that not all potential combinations of parameters were tested and thus do not support the claim that scCLEAN always increases the optimal bottleneck width. The methods outlined above are solely meant to illustrate the distinction in dimensionality

reduction performance between scCLEAN and 10x-V3 and should not be used for downstream biological interpretation.

### PBMC secondary analysis
**Data QC and semi-supervised clustering.** Cell clustering was performed with Seurat v.4.1.1 R toolkit[75]. Genes expressed in less than 3 cells were filtered from the data. Additionally, individual cells with counts less than 200 genes and 500 total UMIs were removed. "scDblFinder" was used for the identification of heterotypic doublets using a cluster-based generation of artificial doublets[76]. The distribution of fraction of MT (mitochondrial) genes was examined and cells within the top 5th percentile were removed. The "SCTransform" function was performed to normalize the count data prior to principal component analysis[77]. We used a residual variance cutoff of 1.3 to calculate the optimal number of variable features. The "RunPCA" function was used for principal component analysis using default settings and the first 50 PCs. The "FindNeighbors" and "RunUMAP" functions were calculated on the first 30 dimensions. The resulting shared-nearest neighbor network graph was used for subsequent k-nearest neighbor clustering. Clusters were identified using the "FindClusters" function using the louvain algorithm with multilevel refinement and resolution set to 1.2. This resolution was used to first over-cluster the data for subsequent doublet removal.

**Reference mapping and cell-type identification.** Query-to-reference mapping was performed by projecting the experimental query datasets (10x-v3 or scCLEAN) onto the curated PBMC atlas, Azimuth using the R package scmap v1.18.0[78]. First, the 500 most informative genes (ranked by variance) were selected through the intersection of variable genes from each individual query and reference datasets. The reference was cluster-indexed by calculating the mean gene expression of the top 500 genes within each annotated cell-type using the indexCluster command. Each query dataset was projected onto the reference using scmapCluster with the similarity threshold = 0.5. Cells with a probability less than the threshold of 0.5 were annotated as "unassigned" while cells with a probability above the threshold were assigned a cell-type label from the reference. Clusters were annotated based on "majority rules" principle from the cell-type labels (>50%) for each cluster. If a cluster had a 1:1 ratio of two distinct cell-type labels, then that individual cluster was divided into sub-clusters using the FindSubCluster command from Seurat and the query dataset was re-mapped onto the reference as described above. The scmap projection for each dataset was assessed for classification accuracy. Cells were then assigned a binary classifier on the agreement between the cell-level annotation and the cluster-level annotation. After binary classification, a generalized linear model (GLM) was fitted using the glm command with the parameters formula = binary classification ~ probability and family = binomial. Using the R package pROC v1.18.0[79].

### Orthogonal validation
**Random matrix theory—signal to noise.** For each sample, random matrix theory was used to distinguish between signal and noise[80], and consequently compare between scCLEAN and 10x-V3. The python tool randomly (https://github.com/RabadanLab/randomly) was used with minimal filtering to accurately characterize the effect of scCLEAN on the signal to noise ratio. Data processing followed the randomly tutorial, initializing the randomly.Rm() model, and using the function preprocess with the following arguments: min_tp = 1, min_genes_per_cell = 1, min_cells_per_gene = 1, refined = True. The model was then refined using the command refining with the flag min_trans_per_gene = 1.

**Deep learning unsupervised clustering.** To ensure that the identification of additional cell clusters with scCLEAN was not an artifact of hyperparameter tuning, deep embedding for single-cell clustering (DESC) (https://github.com/eleozzr/desc) was used, leveraging deep

learning to identify cell clusters in an unsupervised fashion, and assign a confidence score of assigning each cell to a specific cluster following similar initial filtering described above[36]. Despite being an unsupervised method, the proper clustering resolution still had to be manually chosen, so to compare DESC outputs between scCLEAN and 10x-V3, a range of louvain resolutions were selected (0.1, 0.2, 0.4, 0.6, 0.8, 1.0, 1.2, 1.4, 1.6, 1.8, 2.0). For each resolution x, the model was trained using the function train with the following arguments: dims = [anndata.shape[1], 128, 32] tol = 0.001, n_neighbors = 10, batch_size = 256, louvain_resolution = x, do_tsne = True, learning_rate = 300, do_umap = True, num_Cores_tsne = 4.

## Single-cell MAS-Seq

PBMC samples were isolated from a donor and prepared using 10X Genomics Chromium Next GEM Single Cell 3′ Reagent Kit (v3.1) protocol (10X Genomics, CG000315 Rev C) as described above. 100 ng input of amplified GEM barcoded cDNA was then treated with scCLEAN as detailed above with a slight modification. After incubation for 1 h at 37 °C, 1 µL thermolabile proteinase K (New England Biolabs, Ipswich, MA) was added and incubated for 15 min at 65 °C. After scCLEAN and proteinase K treatment, the cleaved targeted sequences were 1X magnetic bead-based size selected using the ProNex beads (Pacific Biosciences, Menlo Park, CA. The remaining desired un-cleaved cDNA samples were PCR re-amplified alongside standard untreated cDNA samples from the same GEM well. The amplified cDNA samples were then 0.95X ProNex size selected for desired <2 kb short transcripts.

Single-cell cDNA fragment sizes were validated post-depletion using the Agilent Bioanalyzer 2100. SMRTbell libraries were generated following the "Single-Cell Iso-Seq SMRTbell Express Template Prep Kit 2.0 Procedure" as per the manufacturer's protocol (Pacific Biosciences PN 101-892-000 v1). Each library was sequenced individually on the PacBio Sequel II System targeting ~3 million full-length single-cell transcript reads per SMRT Cell 8 M. Reads were processed using pigeon (https://github.com/PacificBiosciences/pigeon), the PacBio Transcript Toolkit, following manufacturer's documentation. Seurat compatible inputs were generated using the final, filtered isoforms (containing all genes) using the pigeon make-seurat command.

## MAS-Seq cell QC and downstream analysis

The cells processed with the MAS-Seq protocol and those processed with the addition of scCLEAN workflow were done in parallel using the same parameters for all cell QC and secondary analysis. The cell QC was performed similarly as above with the following modifications. First, genes expressed in less than 3 cells were filtered from the data. By default, when evoking pigeon make-seurat, mitochondrial and ribosomal genes are excluded from further downstream analysis. Thus, to remove low-quality cells, we first plotted the log10(genes) vs log10(UMIs). Cells with both a gene and UMI count depth below 3 median absolute deviations were filtered from the data. To calculate cell complexity, a linear regression model was calculated using the R stats package v4.2.0 using the command lm with formula = log10(genes) ~ log10(UMIs). Model residuals were then calculated using the residuals command with default settings. Cells with residuals less than −0.2 were filtered from the data. Following low-quality cell filtering, predicted doublets were also removed using the doublet detection R package scDblFinder as detailed above.

## Cardiovascular samples

**Cell culture and single-cell isolation.** Human coronary and pulmonary artery smooth muscle cells from four healthy donors were purchased from Cell Applications (San Diego, CA). Cells were donor matched and cultured in growth medium (Cell Applications) at 37 °C with 5% $CO_2$ atmosphere in T25 flasks (Corning, Corning, NY). After 2 passages, cells were detached from cell culture flasks using Trypsin (Invitrogen, Waltham, MA) while incubating at 37 °C for 10 min. Detached cells were washed with PBS and assessed for viability using trypan blue staining (Invitrogen, Waltham, MA) with the Countess II (Invitrogen). Samples were confirmed to have cell viability >90% and were further processed with scRNAseq and scCLEAN as detailed above.

**Trajectory inference.** Cell Rank (https://github.com/theislab/cellrank)[50] was used to calculate dynamic changes in cell states towards terminal lineages. The CytoTRACEkernel was used to generate cell differentiation pseudotime[52]. The transition matrix was calculated using the function "compute_transition_matrix" with the flags threshold_scheme = "soft" and nu = 0.3. To compute terminal states, and the associated fate probabilities, the Generalized Perron Cluster Cluster Analysis estimator was utilized[53]. To identify the top terminal states, schur decomposition was performed using the function compute_schur with n_components = 25 and alpha = 0.2. For both VSMC sample conditions, an eigengap was detected after 2 eigenvalues, so two macrostates (terminal states) were computed using compute_macrostates with n_states = 2. Only the macrostates that had non-zero stationary distance (stationary distribution of coarse-grained transition matrix) were carried forward in analysis. Terminal states were established from the two macrostates, and fate probabilities were identified using compute_absorption_probabilities with time_to_absorption = "all". The top genes associated with each terminal state were identified using the function compute_lineage_drivers.

scFATES (0.8.0) (https://github.com/LouisFaure/scFates) was then utilized to characterize gene-lineage associations with respect to pseudotime, enabling the distinction between genes that drove the split between the two lineages as well as genes highly correlated with each[81]. The tutorial "Conversion from CellRank pipeline" was closely followed. The cell rank analysis was converted to a lineage tree using cellrank_to_tree with the flags time = "pseudotime", Nodes = 10, and seed = 10. The single tip of the tree that was identified closest to the cells of earliest pseudotime was established as the root. scFATES principal graph was calculated using tl.pseudotime with n_map = 100 and seed = 42. The dendrogram was established using the function tl.dendrogram with default parameters. Genes significantly associated to each section of the principal tree were calculated using test_association and fitted to the trajectory using tl.fit. Genes associated with the Coronary and Pulmonary branch were identified using tl.test_fork with the root_milestones flag set to the "Coronary Root Cells" and the milestones flag set to the two terminal branches "Coronary Lineage" and "Late Pulmonary Lineage." Markers were selected using the function branch_specific with the same root and milestones selected as above with effect = 0.3. 91 genes were associated with the Late Pulmonary Lineage and 32 genes were associated with the Coronary Lineage. To identify gene's specific to each module (lineage) at the point of bifurcation with respect to pseudotime inclusion, the function tl.module_inclusion was used with the same root and milestones selected as above in addition to n_map = 50 and parallel_mode = "mappings."

## Statistics and reproducibility

All data were used in the analysis of the presented data with no data excluded from the analysis. Sample size was determined based on conventions in single cell sequencing experiments and as per availability of samples. The investigators were blinded to all identifiable data pertaining to the human cells (PBMCs or VSMCs). Biological and/or technical replicates were performed as appropriate. All samples were prepared for for both traditional and scCLEAN analysis.

## Reporting summary

Further information on research design is available in the Nature Portfolio Reporting Summary linked to this article.

## Data availability

All single-cell sequencing reads, count matrices, and post-processed files are available at the Gene Expression Omnibus (Accession number: GSE283554 and Bioproject PRJNA1194517). Additional processed files are at DOI: 10.6084/m9.figshare.26444173[82]. Publicly available data used for the survey of 14 tissue types for assessment of abundantly expressed transcripts are detailed in Supplementary Data 1.

## Code availability

All code is available at https://github.com/acpandey/scCLEAN and available at https://doi.org/10.5281/zenodo.14079211[83].

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

## Acknowledgements

The authors would like to acknowledge the technical assistance of Abboud Hassan in processing of the sequencing data. This study was supported by the National Center for Advancing Translational Sciences, National Institutes of Health, through UL1TR002550 (E.J.T.) as well as the linked award KL2TR002552 (A.C.P.). It was also supported by American Heart Association Career Development Award 20CDA35310187 (A.C.P.). The content is solely the responsibility of the authors and does not necessarily represent the official views of the NIH/AHA.

## Author contributions

A.C.P. designed the study. A.C.P., J.B., T.M., J.S.H., S.H. performed the experiments. A.C.P., J.B., D.D., E.B.K., F.A., A.C., K.S.C., J.D., A.S., S.R., K.B., J.A., L.M.S. contributed to the analysis of sequencing data. J.B., D.D., A.C., T.M., S.H., P.O. contributed to the sequencing of the samples and preprocessing of samples prior to sequencing. A.C.P., A.S., K.B., J.A. supervised the study. A.C.P., J.B., D.D. wrote the manuscript. A.C.P. and E.J.T. acquired the funding. All authors have approved the final version of the manuscript.

## Competing interests

During the course of this project and/or manuscript preparation, J.B., D.D., A.C., K.C., J.D., A.S., S.R., K.B., and J.A. were employees of Jump-code Genomics. All other authors declare no competing interests.
