## [Transparent Peer Review file · Nature Communications]

A CRISPR/Cas9-based enhancement of high-throughput single-cell transcriptomics

Corresponding Author: Dr Amitabh Pandey

Version 0:

Reviewer comments:

Reviewer #1

(Remarks to the Author)

This is a brief review (3rd review in total) of this manuscript. The methodology, using CRISPR/Cas9 to cleave a substantial catalogue of biologically uninformative, highly abundant sequences in (single-cell) libraries, is a noteworthy contribution to the canon of sequencing applications designed to improve the efficiency and cost-effectiveness of transcriptional profiling. I am satisfied with the changes the authors have made in this version and have opted not to provide any more specific suggestions.

Response to the Reviewers' Critiques:

We thank the Reviewers and Editors for the opportunity to address questions and critiques for our manuscript. We appreciate the insights provided and have made all attempts to incorporate the suggestions from the Reviewers into the revised manuscript. We below have approached the response in a point-by-point manner for each Reviewer.

Reviewer #1 (Remarks to the Author): This is a brief review (3rd review in total) of this manuscript. The methodology, using CRISPR/Cas9 to cleave a substantial catalogue of biologically uninformative, highly abundant sequences in (single-cell) libraries, is a noteworthy contribution to the canon of sequencing applications designed to improve the efficiency and cost-effectiveness of transcriptional profiling. I am satisfied with the changes the authors have made in this version and have opted not to provide any more specific suggestions.

We greatly appreciate the Reviewer's kind comments regarding our manuscript and overall appreciation for our presented work.